# A tool for assessing alignment of biomedical data repositories with open, FAIR, citation and trustworthy principles

**Fiona Murphy** [1] , **Michael Bar-Sinai** [2] , **Maryann E. Martone** [3] *

**1** MoreBrains Cooperative Ltd, Chichester, United Kingdom, **2** Department of Computer Science, Ben-Gurion University of the Negev and The Institute of Quantitative Social Science at Harvard University, Beersheba, Israel, **3** Department of Neurosciences, SciCrunch, Inc., University of California, San Diego, California, United States of America

☯ These authors contributed equally to this work.
* mmartone@ucsd.edu

**Data Availability Statement:** The data outputs and completed questionnaires from the interview analysis are in Zenodo (RRID:SCR_004129): https://zenodo.org/record/4069364. The latest

## Abstract

Increasing attention is being paid to the operation of biomedical data repositories in light of efforts to improve how scientific data is handled and made available for the long term. Multiple groups have produced recommendations for functions that biomedical repositories should support, with many using requirements of the FAIR data principles as guidelines. However, FAIR is but one set of principles that has arisen out of the open science community. They are joined by principles governing open science, data citation and trustworthiness, all of which are important aspects for biomedical data repositories to support. Together, these define a framework for data repositories that we call OFCT: Open, FAIR, Citable and Trustworthy. Here we developed an instrument using the open source Policy-Models toolkit that attempts to operationalize key aspects of OFCT principles and piloted the instrument by evaluating eight biomedical community repositories listed by the NIDDK Information Network (dkNET.org). Repositories included both specialist repositories that focused on a particular data type or domain, in this case diabetes and metabolomics, and generalist repositories that accept all data types and domains. The goal of this work was both to obtain a sense of how much the design of current biomedical data repositories align with these principles and to augment the dkNET listing with additional information that may be important to investigators trying to choose a repository, e.g., does the repository fully support data citation? The evaluation was performed from March to November 2020 through inspection of documentation and interaction with the sites by the authors. Overall, although there was little explicit acknowledgement of any of the OFCT principles in our sample, the majority of repositories provided at least some support for their tenets.

## Introduction

Best practices emerging from the open science movement emphasize that for data to be effectively shared, they are to be treated as works of scholarship that can be reliably found, accessed,

version of the dkNET evaluation instrument is available at http://trees.scicrunch.io/models/dkNET-DRP/start and is made available under a CC-BY 4.0 license. The version used for this study, V1.0 is available at: http://trees.scicrunch.io/models/dkNET-DRP/7/?localizationName=en-US. A copy of the codebook for the instrument along with the visualizations produced by the PolicyModels software is available through Zenodo at (Martone, Murphy, and Bar-Sinai 2020) Additional explication of the Policy Models dimension usage: https://github.com/codeworth-gh/dkNET-DecisionTrees/blob/master/data-repo-compliance/dimension-usage.adoc A snapshot of the code underlying this study is available at: https://doi.org/10.5281/zenodo.4275004 PolicyModels is managed in GitHub (https://github.com/IQSS/DataTaggingLibrary) under an Apache v2 Open-source license. The summary tools (https://github.com/michbarsinai/PolicyModelsSummary) are released under an MIT license.

**Funding:** This work was supported by NIH grant# 3U24DK097771-08S1 from the National Institutes of Diabetes and Digestive and Kidney Diseases. The original decision tree was developed through the FORCE11 Scholarly Commons working group supported by an award from The Leona M. and Harry B. Helmsley Charitable Trust Biomedical Research Infrastructure Program to FORCE11. The authors wish to thank Drs. Ko Wei Lin and Jeffrey Grethe for helpful comments. Fiona Murphy is employed by MoreBrains Cooperative Ltd, Chichester, UK, a commercial company that did not provide any direct funding for this project and does not have any direct interests in the repositories, dkNET or NIH. The funder provided support to UCSD for this work which was paid in the form of salaries for MM and subcontracts to FM and MBS, but did not have any additional role in the study design, data collection and analysis, decision to publish, or preparation of the manuscript. The specific roles of these authors are articulated in the 'author contributions 'section.

**Competing interests:** Dr. Martone is on the board and has equity interest in SciCrunch Inc., a tech startup that develops tools and services in support of Research Resource Identifiers. Dr Murphy is on the board of Dryad Data Repository. These affiliations do not alter our adherence to PLOS ONE policies on sharing data and materials, all of which are shared under an open non-restrictive license.

reused and credited. To achieve these functions, the open science movement has recommended that researchers formally publish their data by submitting them to a data repository [1], which assumes stewardship of the data and ensures that data are made FAIR: Findable, Accessible, Interoperable and Reusable [2]. Publishing data can therefore be seen as equivalent to publishing narrative works in that the locus of responsibility for stewardship transfers from the researcher to other entities, who ensure consistent metadata, future-friendly formats, stable and reliable access, long term availability, indexing and tools for crediting the contributors. As these types of responsibilities are traditionally supported by journals and libraries, it is not surprising that many publishers and libraries are now developing platforms for hosting research data. At the same time, data are not exactly the same as narrative works. They require additional functionality to increase their utility, which explains why the most well known scientific data repositories are led by individual researchers or research communities. Scientific data repositories such as the Protein Data Bank [3] predated the internet and are viewed as important infrastructures for data harmonization, integration and computation.

Although there is general agreement that repositories should support FAIR data, there have been several other community-led initiatives to develop principles in support of open science and data sharing. The "Defining the Scholarly Commons" project at FORCE 11.org identified over 100 sets of principles issued by organizations and groups around the world that cover a range of activities involved in scholarship and how it should be conducted in the 21st century [4]. Common threads included: 1) the need to include not only narrative works, but data, code and workflows; 2) the desire to make these products "as open as possible; as closed as necessary"; 3) FAIRness, i.e., designing the products of scholarship so that they operate efficiently in a digital medium; 4) Citability, i.e., expanding our current citation systems to cover other research outputs like data, and 5) Trustworthiness, i.e., ensuring that those who assume responsibility for stewardship of scholarly output operate in the best interests of scholarship. FORCE11 conducted workshops and exercises to define what a system of scholarly communication should look like in the 21st century. One theme that emerged based on the workshops was that in the imagined scholarly commons, data repositories were central players that provided the human and technical infrastructure for publishing research data. Therefore, data repositories themselves should align with principles governing Open, FAIR, Citable and Trustworthy (OFCT) science.

The FORCE11 exercise was hypothetical but over the years, scholarly communications has been moving towards this vision. As sharing of data and code are increasingly expected and in some cases required, more attention is being paid to the infrastructures that host them and the functions they support. As documented by the Neuroscience Information Framework (NIF; neuinfo.org), on-line data repositories are diverse, each with their own custom user interfaces and few standards as to how they should be designed and the functions they should support [5]. With data repositories increasing in importance, groups have been developing recommendations on a basic set of functions that these repositories should support (e.g., [6–11]. Many of these focus on FAIR, e.g., FAIRshake [6] but they are by no means the only criteria. Although there is considerable agreement across all of these lists, e.g., the use of persistent identifiers, each has a slightly different focus and therefore they are not identical. Rather, they reflect priorities arising out of different contexts. In October 2020, the Coalition of Open Access Repositories (COAR) issued a set of recommendations for data repositories that, like FORCE11, built upon openness, FAIR, Data Citation and TRUST principles, reinforcing the view of the FORCE11 working group that together, OFCT provide a framework for scientific data repositories.

In the work presented here, we developed a set of evaluation criteria based on OFCT by selecting community principles that cover each of these dimensions and operationalizing them

**Table 1. Guiding principles for OFCT used in this study to develop the assessment instrument.**

| Principle | Description | Guiding principles/charters |
|---|---|---|
| Open | Research outputs should be as open as possible and as closed as necessary | Open Definition 2.1 [13] |
| FAIR | Research outputs should be designed to be Findable, Accessible, Interoperable and Reusable for humans and computers | FAIR Data Principles [2] |
| Citable | Research outputs should be supported by formal systems of citation for the purposes of provenance and credit. | Joint Declaration of Data Citation Principles (JDDCP) [14]; Software Citation Principles |
| Trustworthy | Data repositories should demonstrate that they are responsible for long term sustainability and access of data entrusted to them | Principles of Open Infrastructures [15]; Core Trust Seal [16] |

(Table 1). Our aim was to conduct a review of data repositories of relevance to a specific domain, diabetes, digestive and kidney diseases. This work was conducted in the context of the NIDDK Information Network (dkNET.org; [12]). dkNET was established in 2012 to help basic and clinical biomedical researchers find scientific resources relevant to diabetes, digestive and kidney diseases (collectively referred to here as "dk"). dkNET is taking an active role in interpreting and facilitating compliance with FAIR on behalf of this community. Part of this effort involves creating tools to help researchers select an appropriate repository for their data. dkNET maintains a curated list of recommended data repositories that cover domains relevant to dk science, extracted from the resource catalog originally developed by the Neuroscience Information Framework (Cachat et al. 2012 [5]), and cross referenced to repositories recommended by major journals and the National Library of Medicine.

The goal of this repository listing is to make it easier for dk researchers to find an appropriate data repository in support of FAIR, open science and current and upcoming NIH mandates (NOT-OD-21-013: Final NIH Policy for D. . .). We therefore developed an instrument to evaluate these repositories against the OFCT principles to add information to the repository listings that might be important to a researcher in satisfying a mandate, e.g., does the repository issue a persistent identifier as per FAIR, or personal preference, e.g., does the repository support data citation. The instrument was developed using the open source PolicyModels toolkit, a software tool for developing decision trees based on specific policies. In this report, we describe the development and design of an OFCT decision tree, our criteria and strategy for evaluating data repositories for compliance and the results of their application to eight biomedical data repositories from the dkNET listing.

## Materials and methods

We developed a set of 31 questions (Table 2) operationalizing the major elements of each of the principles listed in Table 1. We did not attempt to cover all aspects of the principles, but selected those that were relevant for repositories and for which clear criteria could be developed. At the time we conducted this study, the TRUST principles had not yet been issued and so are not included explicitly in our instrument, although much of what is covered in the Core-TrustSeal is relevant to the TRUST principles. The methods used in this study were not pre-registered prior to conducting the study. For a list of abbreviations used in this text, see S2 Table.

The instrument was used to evaluate eight repositories listed by dkNET (RRID: SCR_001606) provided in Table 3. We selected these repositories to represent different data types or different research foci. Excluded from consideration were repositories that required

**Table 2. Questions and properties used for the final interview, the table shows the question order (Q#), the text of the question posed in the interview (Question text), possible answers (Answers), whether or not the question is conditional ("C"), the dependencies of conditional questions (D) and the principle(s) the question is meant to cover (P).**

| Question text | Answers | C | D | P |
|---|---|---|---|---|
| 1. Does the repository provide access to the data with minimal or no restrictions? (acc) | no restrictions minimal restrictions significant restrictions significant but not justified | N | | O |
| 2. Are you free to reuse the data with no or minimal restrictions? (reuse) | yes somewhat no | N | | O |
| 3. Does the repository provide a clear license for reuse of the data? (lic-clr) | dataset level repository level no license | N | | F |
| 4. Are the data covered by a commons-compliant license? (lic-cc) | best good somewhat open closed | Y | #3 | O |
| 5. Does the repository platform make it easy to work with (e.g. download/re-use) the data? (plat) | yes no | N | | F |
| 6. Does the repository require or support documentation that aids in proper (re)-use of the data? (ru-doc) | best good adequate lacking | N | | F |
| 7. Does the repository provide a search facility for the data and metadata? (sch-ui) | yes no | N | | F |
| 8. Does the repository assign globally unique and persistent identifiers (PIDs)? (pid-g) | yes no | N | | F |
| 9. Does the repository allow you to associate your ORCID ID with a dataset? (orcid) | required supported not available | N | | C |
| 10. Does the repository support the addition of rich metadata to promote search and reuse of data? (md-level) | rich limited minimal | N | | F |
| 11. Are the (meta)data associated with detailed provenance? (md-prov) | best good worst | N | | F |
| 12. Does the repository provide the required metadata for supporting data citation? (md-daci) | full partial none | N | | C |
| 13. Do the metadata include qualified references to other (meta)data? (md-ref) | best good worst | N | | F |
| 14. Does the repository support bidirectional linkages between related objects such that a user accessing one object would know that there is a relationship to another object? (md-lnk) | best good unclear worst | N | | F |
| 15. Does the repository enforce or allow the use of community standards for data format or metadata? (fmt-com) | yes no | N | | F |
| 16. Does the repository accept metadata that is applicable to the dkNET community disciplines? (md-dkn) | best good worst | N | | F |
| 17. Does the repository have a policy that ensures the metadata (landing page) will persist even if the data are no longer available? (md-psst) | no by evidence by policy | N | | F |

*(Continued)*

**Table 2.** (Continued)

| Question text | Answers | C | D | P |
|---|---|---|---|---|
| **18. Do the metadata use vocabularies that follow FAIR principles? (md-FAIR)** | enforced<br>allowed<br>minimal | N | | F |
| **19. Does the machine-readable landing page support data Citation? (land-ctsp)** | yes<br>no | N | | C |
| **20. Does the repository use a recognized community standard for representing basic metadata? (md-cs)** | yes<br>no | N | | F |
| **21. Can the (meta)data be accessed via a standards compliant API? (acc-api)** | yes<br>no | N | | F |
| **22. Do the metadata use a formal accessible shared and broadly applicable language for knowledge representation? (md-vcb)** | yes<br>no | N | | F |
| **23. Does the repository provide API-based search of the data and metadata? (sch-api)** | yes<br>no | N | | F |
| **24. Is the governance of the repository transparent? (gov-tsp)** | best<br>good<br>worst | N | | T |
| **25. Is the code that runs the data infrastructure covered under an open source license? (oss)** | best<br>good<br>no | N | | T |
| **26. Has the repository been certified by Data Seal of Approval or the Core Trust Seal or equivalent? (tr-seal)** | yes<br>no | N | | T |
| **27. Is the repository stakeholder governed? (gov-stk)** | full<br>good<br>weak<br>none | N | | T |
| **28. Does the repository provide a machine-readable landing page? (land-api)** | yes<br>no | Y | #29 | F |
| **29. Does the PID or other dataset identifier resolve to a landing page that describes the data? (land-pg)** | yes<br>no | Y | #8 | C |
| **30. Does the metadata clearly and explicitly include identifiers of the data it describes? (md-pid)** | all<br>some<br>none | Y | #8,<br>#29 | F |
| **31. Does the repository assign, or the contributor provides, a locally unique identifier to the dataset or the data contribution? (pid-l)** | yes<br>no | Y | | F |

A "Y" in the conditional column indicates that whether or not the question is shown to the interviewer depends upon a prior answer. The questions that elicit the conditional questions are shown in the Dependencies column. Each question is assigned a unique ID which is shown in parentheses after each question. Y = Yes, N = No, O = Open, F = FAIR, C = Citable, T = Trustworthy. The full instrument, which also includes explanatory text and appropriate links, is available at [17].

an approved account to access the data, e.g., the NIDDK Central Repositories. We also did not consider knowledge bases, defined here as a database that extracts observations from the literature or as a result of analyses of primary data, but not the primary data themselves. We did, however, include AMP-T2D which presents statistical summaries of clinical data although it does not host the primary data. We also excluded some of the most well known of the biomedical databases, e.g., the Protein Data Bank and GEO, in order to focus on dk-relevant but perhaps lesser known repositories. We included two generalist repositories, Zenodo and NIH-Figshare, as the generalist repositories are likely to play a significant role for diverse domains like dk, where specialist repositories for all data types and research foci may not be available. NIH-Figshare at the time of evaluation was made available as a pilot by the National Library of Medicine for data deposition by NIH-supported researchers. Many of these

**Table 3. List of repositories evaluated in this study.**

| Repository | Description | Section | URL |
|---|---|---|---|
| Accelerating Medicine Partnership Type 2 Diabetes (RRID:SCR_003743) | Portal and database of DNA sequence, functional and epigenomic information, and clinical data from studies on type 2 diabetes and analytic tools to analyze these data. | T2DKP Datasets under "Data" tab | http://www.kp4cd.org/datasets/t2d |
| Cell Image Library (RRID: SCR_003510) | Freely accessible, public repository of vetted and annotated microscopic images, videos, and animations of cells from a variety of organisms, showcasing cell architecture, intracellular functionalities, and both normal and abnormal processes. | Main site representing single image and datasets | http://www.cellimagelibrary.org |
| Flow Repository (RRID: SCR_013779) | A database of flow cytometry experiments where users can query and download data collected and annotated according to the MIFlowCyt data standard. | Public site | http://flowrepository.org |
| Image Data Resource (IDR) (RRID:SCR_017421) | Public repository of reference image datasets from published scientific studies. IDR enables access, search and analysis of these highly annotated datasets. | Cell-IDR | http://idr.openmicroscopy.org/cell/ |
| Mass Spectrometry Interactive Virtual Environment (MassIVE) (RRID: SCR_013665) | MassIVE is a community resource developed by the NIH-funded Center for Computational Mass Spectrometry to promote the global, free exchange of mass spectrometry data. | Access public datasets | https://massive.ucsd.edu/ProteoSAFe/datasets.jsp#%7B%22query%22%3A%7B%7D%2C%22table_sort_history%22%3A%22createdMillis_dsc%22%7D |
| Metabolomics Workbench (RRID:SCR_013794) | Repository for metabolomics data and metadata which provides analysis tools and access to various resources. NIH grantees may upload data and general users can search metabolomics database. | Data Repository | https://www.metabolomicsworkbench.org/data |
| NIH Figshare (RRID: SCR_017580) | Repository to make datasets resulting from NIH funded research more accessible, citable, shareable, and discoverable. | Public portal and password protected space | https://nih.figshare.com/ |
| Zenodo (RRID:SCR_004129) | Repository for all research outputs from across all fields of science in any file format. | Public site and data submission forms | https://zenodo.org/ |

The specific section of the repository evaluated is indicated in the Section column, along with the corresponding URL.

repositories are complex websites with multiple tools, services and databases, and so for each of the repositories, we indicate in Table 3 which specific component(s) were reviewed.

## Developing and testing the instrument

To design the instrument, we adapted the decision tree originally designed by the FORCE11 Scholarly Commons project for evaluating repositories on OFCT principles [4]. We benchmarked the instrument against a range of surveys and other tools then available for similar uses. The process involved reading the background rationale and information about these other instruments, and assessing them for their objectives, projected user profiles, scope and outputs. Factors such as: is this publisher/publication facing; is it community-led or commercial; if referring to, say, repositories, is it intended to inform potential users about the repositories, or to support repository managers themselves? The materials in question included the repository finder tool developed by DataCite for the Enabling FAIR Data project; the Scientific Data journal repository questionnaire; the FAIRsFAIR data assessment tool; and the Core Trustworthy Data Requirements. From this exercise, we determined that the answers to the questions were sometimes difficult to ascertain as clear criteria for evaluation had not been specified. Some areas were clearly missing while some of the questions were duplicative. We thus modified the questionnaire by removing duplicates, adding additional questions,

developing specific evaluation criteria and adding tips as to where to look for certain types of information. Definitions and links to supporting materials were also provided for each question where appropriate. The complete version of the questionnaire used here, which includes the criteria used for each question, was deposited in Zenodo [17].

The final questionnaire comprised 31 questions, listed in order in Table 2. Some of the questions are conditional, that is, their presentation is dependent upon a prior answer. For example, if an interviewer answered "No" to question #3 "Does the repository provide a clear license for reuse of the data?" then question #4 "Are the data covered by a commons-compliant [i.e., open] license?" is not presented. Thus, the total number of questions asked may differ across repositories. Each of the questions was given a unique ID which is supplied after the question in Table 2 can be found organized alphabetically in S1 Table.

Table 2 also lists the principle set it covers (OFCT). Although the questions were originally grouped by principle, when testing the questionnaire we noted that many questions were logically related to one another, e.g., under the FAIR section we asked about licenses, while under the open section we asked about open licenses. Therefore, we reordered the questions to reflect better the actual workflow a reviewer might implement by grouping together related questions.

**Encoding the instrument in policy models.** The questionnaire was encoded using the PolicyModels software (RRID:SCR_019084). PolicyModels uses formal modeling to help humans interactively assess artifacts or situations against a set of rules. A PolicyModels model consists of an n-dimensional space (called "policy space"), and a decision graph that guides users through that space using questions. Each of the policy space's dimensions describes a single assessed aspect using ordinal values. Thus, every location in a policy space describes a single, discrete situation with regards to the modeled guidelines [18].

The dimensions of the policy space defined for this work formally capture the assessment aspects implied by OFCT. It contains 45 dimensions that are assessed by the 31 questions shown in Table 2, such as Documentation Level (lacking/adequate/good/full), Metadata Provenance (unclear/adequate/full), and overall ratings of each criteria, e.g., FAIR Accessibility level (none/partial/full) and so forth. The full policy space for this instrument is shown in Fig 1, and is also available via the questionnaire landing page and in [17]. Some dimensions are assigned based on the answer to a single question, while some are calculated based on values on other dimensions. Using an interactive interview guided by our model's decision graph, we were able to find the location of each of the evaluated repositories in the space we defined. To visualize this space, we developed an interactive viewer available at http://mbarsinai.com/viz/dknet. This allowed us to formally compare repositories across multiple dimensions, and to collect overall statistics.

The main features of the tool are shown in Fig 2. The online version allows interviewers to annotate the response to each question with notes (Fig 2B) and export the outcomes of the evaluation (Fig 2G). Currently, the results can only be exported as.json or.xml. However, to save a human readable version.pdf version of the questionnaire results, users can use the browser's print function to save the interview summary page as a PDF.

## Scoring

Five of the sites were reviewed independently by FM and MM between March and May 2020 and three in December 2020. Results were compared and a final score assigned for each question. The reviewers made a good faith effort to find information on the site to provide an accurate answer for each question. The evaluation included checking of information on the repository site, examination of the metadata provided by the site, investigations into the PID

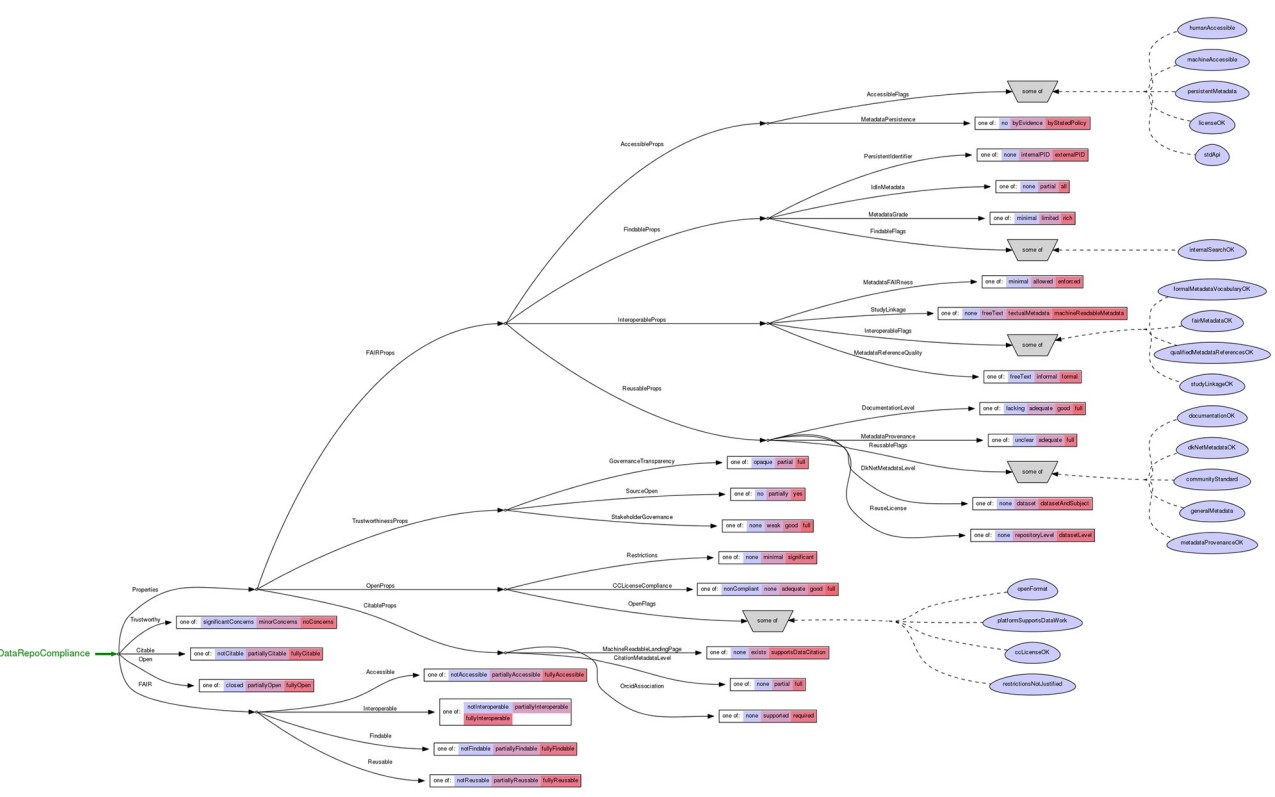

**Fig 1. Policy space defined by the PolicyModels software illustrating the relationship of the dimensions assessed to the properties (rectangles) and flags (blue ovals).** A full resolution view is available in Martone et al., (2020) [17].

system, including what information was exported to DataCite if DOIs were used, inspection of the underlying platform code, documentation and tutorials. For some of the repositories, we created accounts in order to evaluate practices and further documentation for uploading data, e.g, can one associate an ORCID with a dataset, although in no case did we actually upload any data. To check machine-readability for data citation, we attempted to import the citation metadata into an on-line reference manager to see if they were recognized. We did not attempt to read papers that described the site. If we could not find explicit evidence for a criterion, we assumed that it was not present. Therefore, a "No" answer to a question such as "Does the repository provide an API" could mean either that the repository has a statement saying that it will not provide an API, or that we could find no evidence that it did. After the study was completed, we sent a copy of the assessments to the owner or contact for each of the 8 repositories asking them to review our results for accuracy. We received acknowledgements from 7 and responses from 3 of them. We provide both the original and the corrected versions in [19].

After a model-based interview regarding a given repository is completed, PolicyModels displays a coded evaluation of the repository. Formally, PolicyModels locates the coordinate that best describes that repository in our model's policy space. While mathematically all dimensions are equally important, PolicyModels allows its users to organize them hierarchically, to make working with them more comfortable.

Our proposed model's policy space is organized as follows. High-level property descriptions, such as openness and citability levels, are each represented in a dimension of their own. These dimensions have three levels, corresponding to "not at all", "somewhat", and "fully". For

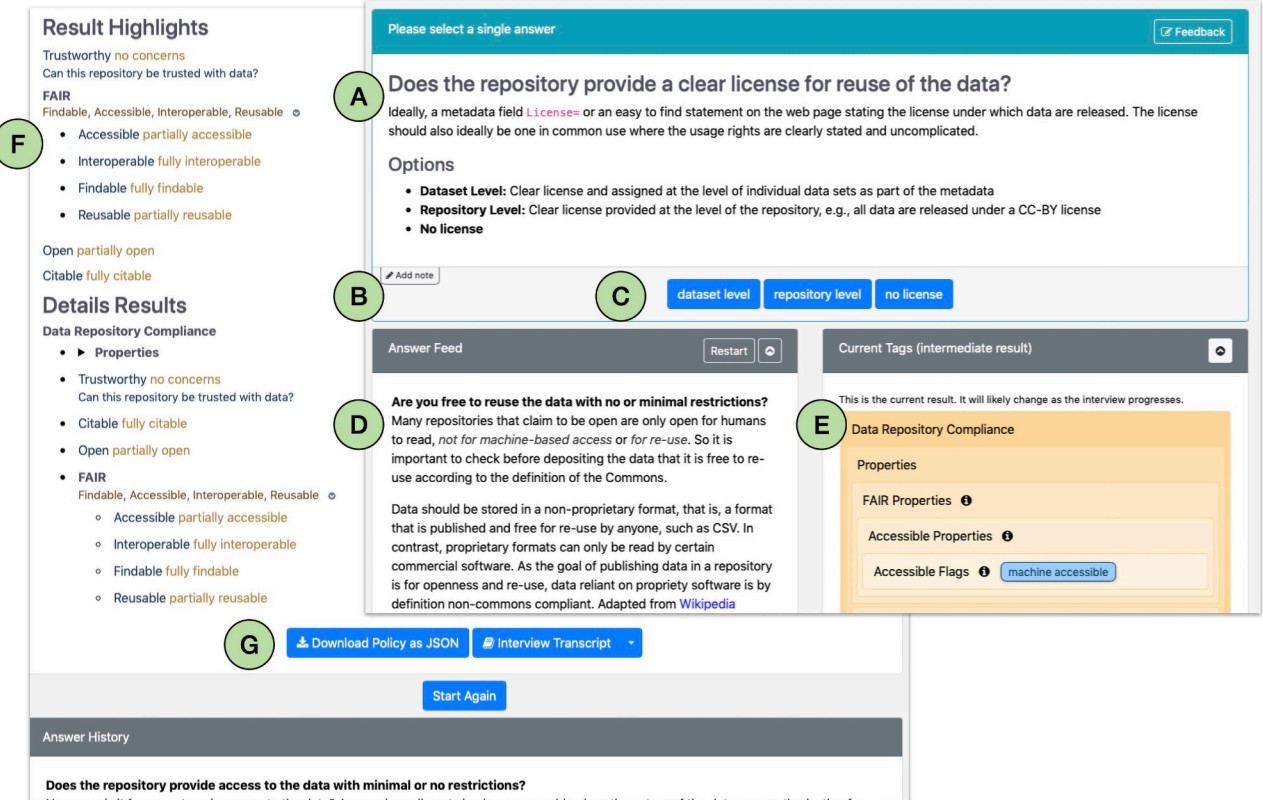

**Fig 2. Main features of policy models questionnaire.** The panel on the right provides an example of the question panel and the left panel shows the results of a survey after it is completed. A) each question is presented in sequence and can be accompanied by explanatory material and links to additional material; B) The interviewer may add notes to each question; C) Interviewer records an answer by selecting the appropriate response; D) The answer feed may be displayed and used to track progress and also to allow an interviewer to revisit a question to change an answer; E) Policy models tallies the answers and assigns tags assessing compliance with OFCT; F) Final tags assigned for each category; G) The results may be downloaded as json or xml.

example, the Reusable dimension contains the levels "not reusable", "partially reusable", and "fully reusable".

The high-level properties are a summary of lower-level assertions, each describing a narrow aspect of these high-level properties. These assertions can be binary or detailed. For example, "open format", one of the openness sub-aspects, is "yes" for repositories that use an open format and "no" for the others. On the other hand, "Study Linkage", an interoperability sub-aspect, can be "none", "free text", "textual metadata", or "machine readable metadata".

Each interview starts by pessimistically setting all high-level dimensions to their lowest possible value: "not at all". During the interview, while lower-level aspect results are collected, high-level repository coordinates may be advanced to their corresponding "somewhat" levels. After the last question, if the evaluated repository achieved an acceptable for all sub-aspects of a certain higher property, that property is advanced to its "fully" level.

As a concrete example, consider the "Findable" dimension. At the interview's start, we set it to "not findable". During the interview, our model collects results about persistent identifiers used by the repository (none/internal/external), the grade of the metadata it uses (minimal/limited/rich), whether ids are stored in the metadata (none/partial/all), and whether the repository offers an internal search feature (yes/no). If a repository achieves the lowest values in all these dimensions, it maintains its "not findable" score. If it achieves at least one non-lowest

value, it is advanced to "partially findable". After the interview is completed, if it achieved the highest value in each of these dimensions, it is advanced to "fully findable".

## Results

### Overall impressions

Fig 3 provides the average score, scaled to a 10 point scale for each question, with 1 = lowest score and 10 = best score. Scaling was performed because each question can have a different

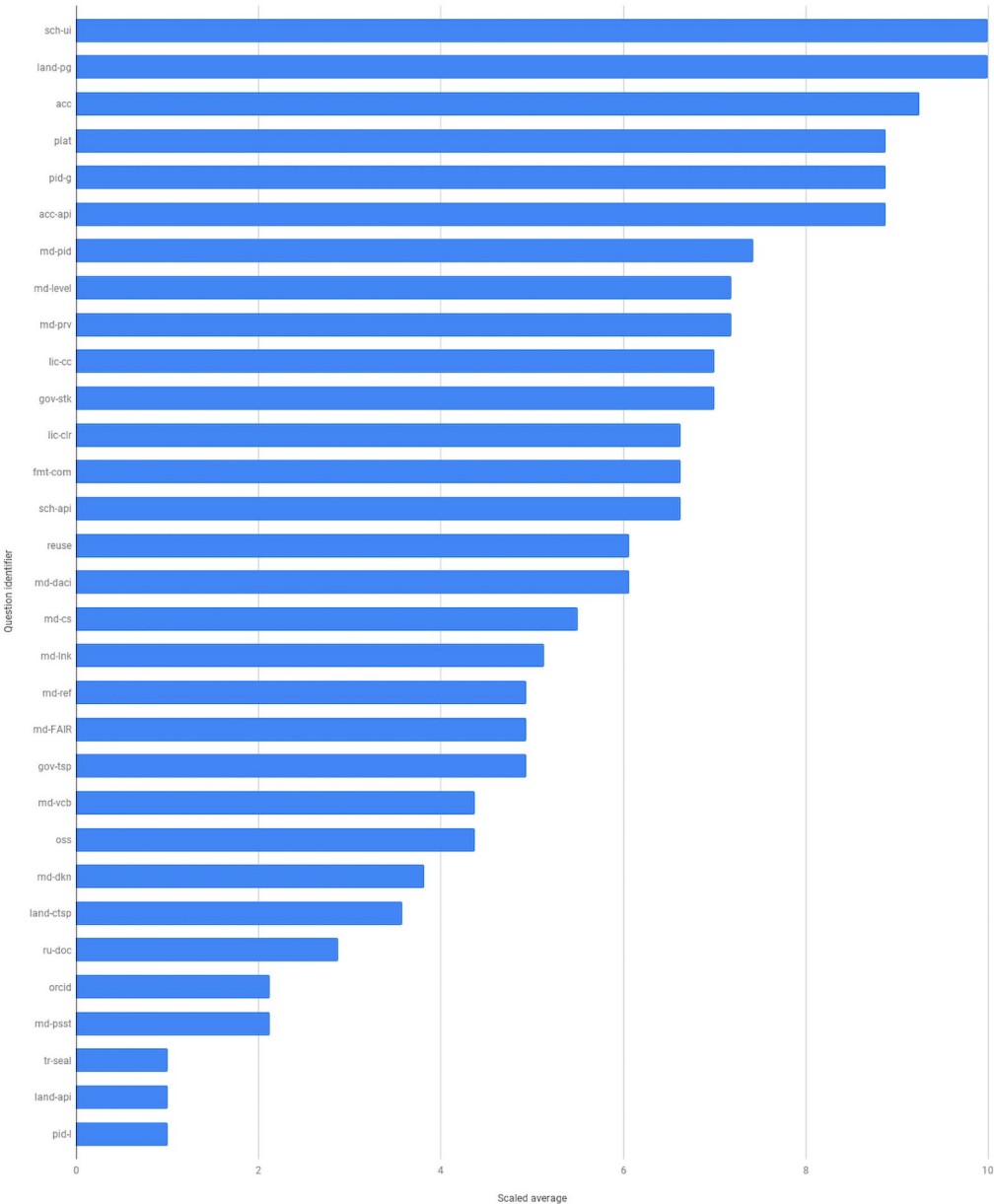

**Fig 3. Average scaled score for each question across all repositories.** Questions are ordered on the Y axis according to highest average score (top) to lowest score (bottom). The data underlying the figure is available in Bar-Sinai et al., 2020 in the summary-transcript.tsv file. The average scaled score was calculated per question and then the results were sorted from highest to lowest. Question ID key is found in S1 Table.

number of answers, making it difficult to compare repositories across questions. A full list of question IDs is available in Supplemental Material S1. On over half of the questions (18/31), repositories scored on average higher than the midpoint, indicating at least some alignment. On under half they were at or below the midline (13/31), indicating poor alignment or no information available, with all repositories receiving the lowest score on 3 of the questions. We note, however, that of the 3 lowest scores, only tr-seal ("Has the repository been certified by Data Seal of Approval or the Core Trust Seal or equivalent") was assessed across all repositories. The other two were conditional questions with only 1 assessment for each.

The answers to these questions are used to assign OFCT properties and flags in the Policy Space. Flags represent a binary rating; if the flag is assigned, then the repository meets that criterion, e.g., openFormat means that the repository makes data available in an open format. Properties are rated on an ordinal scale generally that indicates full, partial or minimal compliance. The properties and the flags assigned by the PolicyModels software and their meaning are provided in Table 4.

Our instrument calculates an overall rating per OFCT dimension, as shown in Fig 4. For a repository to be rated fully compliant, it would have to receive an acceptable score for all dimensions that evaluate that principle; conversely to be rated non-compliant would require an unacceptable score on all dimensions. This calculation is performed using PolicyModels, and is based on the range of acceptable and unacceptable values in various dimensions of the instrument's policy space. Note that we do not provide scores for individual repositories in this paper, as our intent is not to grade them. However, the completed questionnaires for the individual repositories are available in [20].

We sent the completed versions to the repositories for their inspection. We received acknowledgements from 7 of the repositories and responses from 3 of them. Two of the repositories each noted two errors in our evaluation. For one repository, one error was a misinterpretation on our part and one was made because the documentation of that function was not clearly visible on the site. For the other, the errors were on our part. For the third repository, AMP-T2D, we went over our responses with the repository representative in person. This site was difficult to review as it is not a repository for primary data but rather presents harmonized summary results from clinical studies. We agreed that there were 4 errors on our part, and two where there was some disagreement as to whether an aggregator such as AMP-T2D was responsible for such a function. One question was also scored negatively because of lack of relevant documentation on the site. These results suggest that overall, we were reasonably accurate in our evaluation of the repositories. In the following figures, we use the corrected results if we made the error but not if it was a difference of opinion or if the necessary documentation was not present.

As seen in Fig 4, at least one repository scored as fully compliant in each of the Open, Findability, Accessibility, Reusability and Citability dimensions. Conversely, three repositories received the lowest rating for Findability and one for Citability. No single repository was equally good—or bad—on all dimensions, that is, the same repositories did not receive either all of the highest or lowest scores. The most flags assigned to a single repository was 15 while the fewest was 5.

## Open dimension

Seven repositories were scored as "Partially Open" and one as fully open (Fig 4) with details of the policy space for open criteria shown in Table 4. As biomedical repositories can deal with sensitive information that cannot be openly shared, they should adhere to the "As open as possible; as closed as necessary" principle. However, none of the repositories we evaluated had

**Table 4. Ratings for each OFCT property and flag.**

| Properties and flags | Counts | Description |
|---|---|---|
| **Open** | | |
| Restrictions | none:6 minimal:2 significant:0 | Level of restrictions imposed by the repository in order to access datasets. |
| CCLicenseCompliance | nonCompliant:0 none:3 adequate:1 good:3 full:1 | Commons-compliance level of the repository license |
| openFormat | no:4 yes:4 | Is the data available in an open (non-proprietary) format? |
| platformSupportsDataWork | no:1 yes:7 | Does the repository platform make it easy to work with (e.g. download/re-use) the data? |
| ccLicenseOK | no:3 yes:5 | Are the data covered by a commons-compliant license? (any answer except "closed" is considered a "yes") |
| restrictionsNotJustified | no:8 yes:0 | Does the repository impose "significant but not justified restrictions" on accessing the data? |
| **FAIR:Findable** | | |
| PersistentIdentifier | none:1 internalPID:0 externalPID:7 | Scope of persistent identifier assigned to the data, if any |
| IdInMetadata | none:1 partial:2 all:4 | Does the metadata clearly and explicitly include the identifier of the data it describes?} |
| MetadataGrade | minimal:0 limited:5 rich:3 | Level of additional metadata that can be added to promote search and reuse of data |
| FindableFlags/ internalSearchOK | no:0 yes:8 | Does the repository provide a search facility for the data and metadata? |
| **FAIR:Accessible** | | |
| humanAccessible | no:1 yes:7 | Does the repository provide access to the data with minimal or no restrictions? |
| machineAccessible | no:2 yes:6 | Can the data be accessed by a computer? Note that this includes access both via UI and API, as web-based UI is by definition machine-accessible. |
| persistentMetadata | no:7 yes:1 | Does the repository have a policy that ensures the metadata (landing page) will persist even if the data are no longer available, either by policy or example? |
| licenseOK | no:3 yes:5 | Does the repository provide a clear license for reuse of the data? (any answer except "no license") |
| stdApi | no:1 yes:7 | Can the (meta)data be accessed via a standards compliant API? |
| MetadataPersistence | no:7 byEvidence:0 byStatedPolicy:1 | Does the repository have a policy that ensures the metadata (landing page) will persist even if the data are no longer available? |
| **FAIR:Inteoperable** | | |
| MetadataFAIRness | minimal:3 allowed:3 enforced:2 | Do the metadata use vocabularies that follow FAIR principles? |
| StudyLinkage | none:0 freeText:6 textualMetadata:1 machineReadableMetadata:1 | Type of linkage between the published dataset and the paper that accompanied it |
| formalMetadataVocabularyOK | no:5 yes:3 | Do the metadata use a formal, accessible, shared and broadly applicable language for knowledge representation? |
| fairMetadataOK | no:3 yes:5 | Do the metadata use vocabularies that follow FAIR principles? (any answer except "minimal") |
| qualifiedMetadataReferencesOK | no:3 yes:5 | Do the metadata include qualified references to other (meta)data? (any answer except "worst") |
| studyLinkageOK | no:6 yes:2 | Linkage between the published dataset and the paper that accompanied it is "good" or "best". |
| MetadataReferenceQuality | freeText:3 informal:3 formal:2 | Type of qualified references to other (meta)data, included in the (meta)data stored in the repository |
| **FAIR:Reusable** | | |
| DocumentationLevel | lacking:4 adequate:3 good:1 full:0 | Level of support offered by the repository for documentation that aids in proper (re)-use of the data |
| MetadataProvenance | unclear:0 adequate:5 full:3 | Are the (meta)data associated with detailed provenance? |
| documentationOK | no:4 yes:4 | Does the repository require or support documentation that aids in proper (re)-use of the data? (any answer except "worst") |
| dkNetMetadataOK | no:5 yes:3 | Does the repository accept metadata that is applicable to the dkNET community disciplines? (any answer except "worst") |

**Table 4.** (Continued)

| Properties and flags | Counts | Description |
|---|---|---|
| communityStandard | no:3 yes:5 | Does the repository enforce or allow the use of community standards for data format or metadata? |
| generalMetadata | no:4 yes:4 | Does the repository use a recognized community standard for representing basic metadata? |
| metadataProvenanceOK | no:0 yes:8 | Are the (meta)data associated with detailed provenance? (any answer except "worst") |
| DkNetMetadataLevel | none:5 dataset:1 datasetAndSubject:2 | Does the repository accept metadata that is applicable to the dkNET community disciplines? |
| ReuseLicense | none:3 repositoryLevel:0 datasetLevel:5 | Level at which the repository provides a clear license for reuse of the data |
| **Citable** | | |
| MachineReadableLandingPage | none:1 exists:5 supportsDataCitation:2 | Level of machine-readability of the dataset landing page (if any) provided by the repository |
| CitationMetadataLevel | none:2 partial:3 full:3 | Does the repository provide the required metadata for supporting data citation? |
| OrcidAssociation | none:6 supported:2 required:0 | Does the repository allow the authors to associate their ORCID ID with a dataset? |
| **Trustworthy** | | |
| GovernanceTransparency | opaque:2 partial:5 full:1 | Transparency level of the repository governance |
| SourceOpen | no:5 partially:0 yes:3 | Is the code that runs the data infrastructure covered under an open source license? |
| StakeholderGovernance | none:0 weak:2 good:2 full:2 | Level of control stakeholders have in the repository's governance |

Overall ratings on each dimension measuring OFCT. Properties (Props) and Flags are assigned by the PolicyModels software based on the answers given. Properties are assigned at multiple levels depending on level of compliance, whereas all flags are binary and are only assigned if the repository meets the criteria. Repository Count = number of repositories with each rating; QID: ID of question that assigns the property/flag; Short explanation: Meaning of the property or flag.

sensitive data and all were judged to make their data available with minimal to no restrictions, i.e., no approval process for accessing the data. We also evaluated repositories' policies against the open definition: "Knowledge is open if anyone is free to access, use, modify, and share it—subject, at most, to measures that preserve provenance and openness." Thus, data have to be available to anyone, including commercial entities, and users must be free to share them with others. We thus examined the licenses against those rated by the Open Knowledge Foundation as adhering to their definition (https://opendefinition.org/licenses/). One repository was considered fully compliant, 4 were rated as "good" with respect to open licenses, 3 had no licenses (Table 4; CCLicenseCompliance). The four rated as "good" did not receive the best score due to allowing the user to select from a range of licenses, some of which restricted commercial use.

## FAIR dimension

Our questions on FAIR evaluated both compliance with specific FAIR criteria, e.g., the presence of a persistent identifier or with practices that support FAIR, e.g., providing landing pages and providing adequate documentation to promote reuse. Evaluating a repository against some principles also required that we define concepts such as "rich metadata" (FAIR principle F2) and a "plurality of relevant attributes" (FAIR principles R1).

Rich metadata were considered to comprise basic descriptive metadata, i.e., dataset title, description, authors but also metadata specific to biomedical data, e.g., organism, disease conditions studied and techniques employed (Q:**md-level**). "A plurality of relevant attributes" was defined in question **md-dkn** as providing sufficient metadata to understand the necessary context required to interpret a dkNET relevant biomedical dataset. Such metadata includes subject level attributes, e.g., ages, sex and weight along with detailed experimental protocols. Fig 5

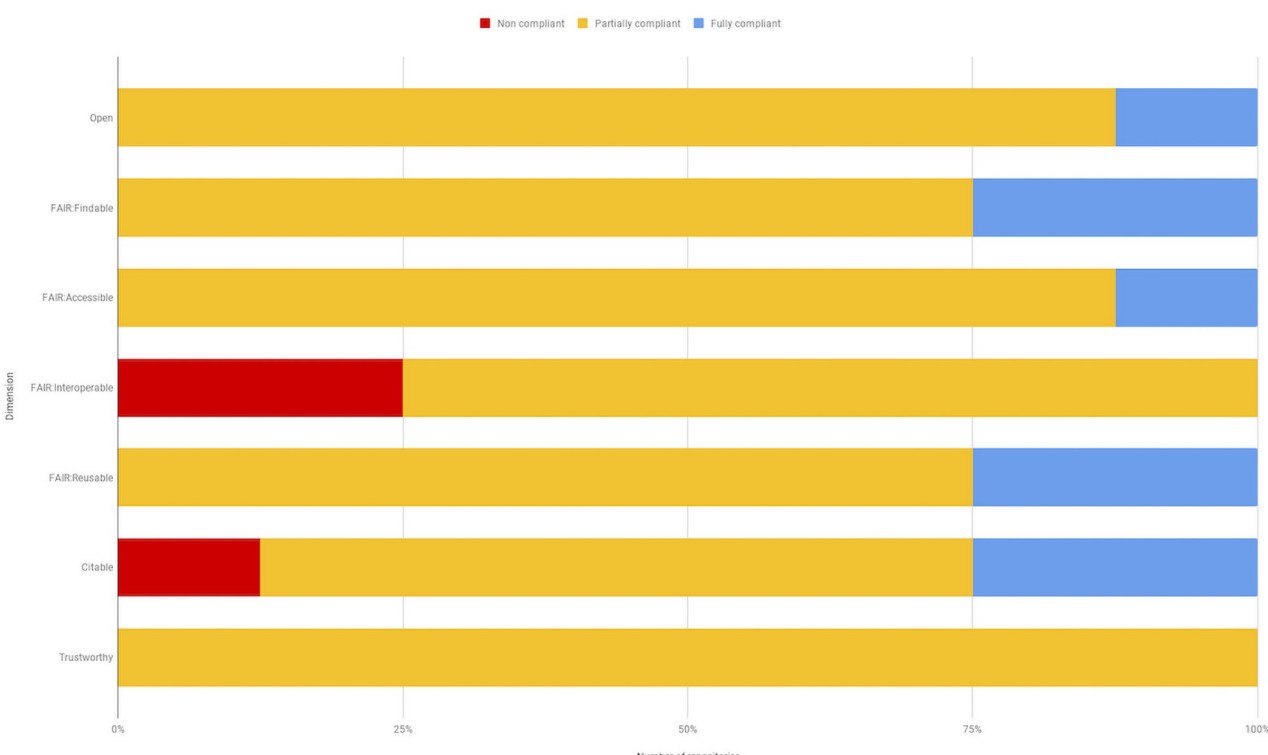

**Fig 4. Overall ratings of repositories on OFCT criteria.** The Y axis shows the individual dimensions and the X axis shows the number of repositories assigned each rating out of the 8 assessed. Red = Not compliant; Gold = Partially compliant; Blue = Fully compliant.

positions each repository in the metadata policy space and shows that only one repository fully satisfied both metadata requirements.

Fig 4 shows that the majority of repositories were either partially or fully compliant with all the Findability and Accessibility dimensions. Two repositories achieved the highest rating in Findability. Seven out of the 8 repositories supported external PIDs, either DOIs or accession numbers registered to identifiers.org. One repository issued no identifiers. Only 1 repository was considered fully accessible because only 1 repository had a clear persistence policy (Q:md-psst). Both the Data Citation and FAIR principles state that metadata should persist even if the accompanying data are removed. We considered either an explicit policy or clear evidence of such a practice as acceptable, e.g., a dataset that had been withdrawn but whose metadata remained.

Overall scores were lowest for the interoperability dimensions, with 3 repositories being judged non-interoperable. Only one of the repositories achieved the StudyLinkage flag which indicated that they had fully qualified references to other data, in other words, that the relationship between a metadata attribute and a value was both machine readable and informative. We measured this property by looking at how repositories handled supporting publications in their metadata, e.g., did they specify the exact relationship between the publication and the dataset? To measure this, we looked at the web page markup ("view source") and also checked records in DataCite.

Two repositories achieved the highest score for reusability, while the remainder were considered partially reusable. Five repositories were judged as having inadequate metadata for providing experimental context, 4 as having inadequate user documentation, while 3 did not provide a clear license.

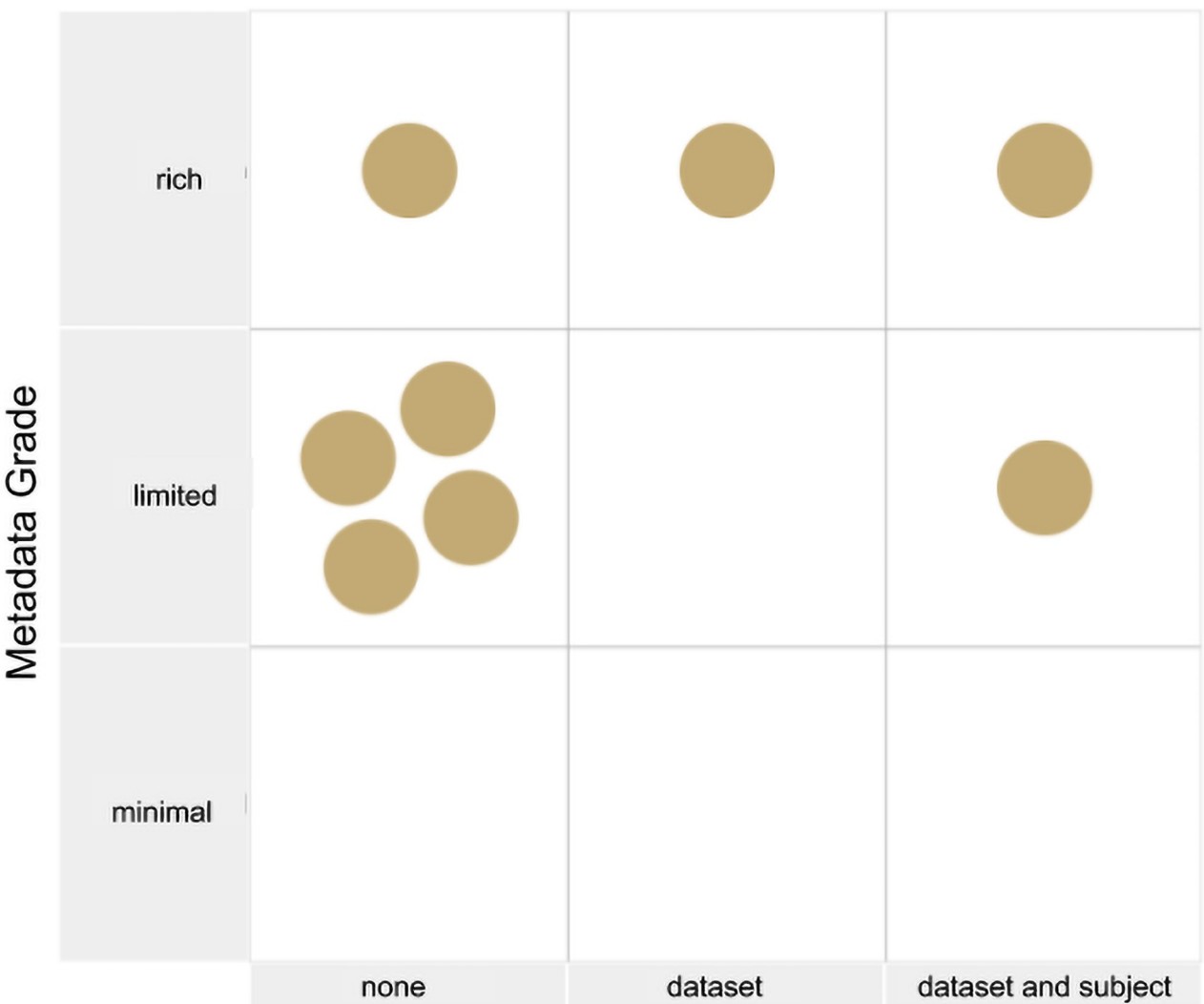

**Fig 5. Assessment of the degree of descriptive metadata (X) vs relevant biomedical metadata (dkNET metadata level) (Y).** The Metadata Grade assesses whether the repository complies with the Findable principle for Rich Metadata, while the dkNET metadata measures the degree to which the repository supports the Reusable principle requiring "a plurality of relevant attributes". Relevance here was assessed with respect to dkNET. Only one repository received the highest score for both categories.

### Citable dimension

Data citation criteria included the availability of full citation metadata and machine-readable citation metadata according to the JDDCP ( [10, 21, 22] ). We also evaluated the use of ORCIDs, as linking ORCIDs to datasets facilitates assigning credit to authors. As shown in Fig 6, only two repositories supported ORCID and provided full citation metadata. Consequently, 2 repositories were judged to fully support data citation, while the remainder were judged as partially (N = 5) or not supporting (N = 1) data citation. Many of the repositories had a citation policy, but most of these policies requested citation of a paper describing the repository and contributor of the data acknowledged rather than creating full citations of a particular dataset.

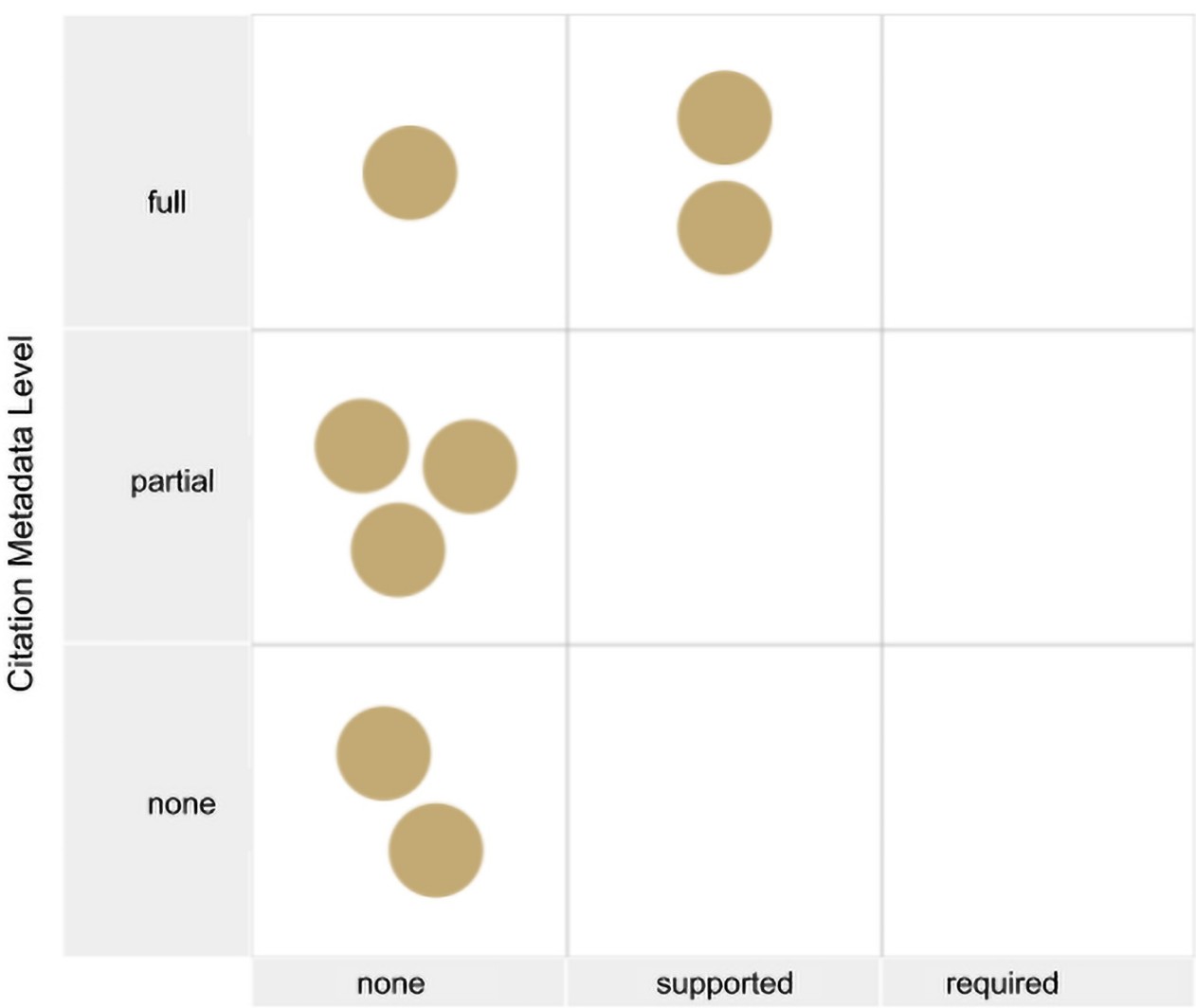

**Fig 6. Repositories plotted against two dimensions of data citation.** The Y axis shows support for citation metadata and the X axis for ORCID support. Two repositories support ORCID and provide full citation metadata. Two repositories have no support for data citation and the others have partial support.

Two were judged not to have sufficient metadata to support full citation, e.g., listing only the submitter and not other authors [see question med-daci].

### Trustworthy dimension

Trustworthiness was largely assessed against the Principles of Open Infrastructures (Bilder et al., 2015) and the CoreTrustSeal criteria. As noted in the introduction, the TRUST principles had not been issued when we developed our instrument and so are not explicitly included. The questionnaire originally probed the different certification criteria recommended by the Core-TrustSeal but we dropped this approach in favor of a single binary question on whether or not the repository was certified by CoreTrustSeal or equivalent. If a repository was certified, it

would automatically be rated fully trustworthy. However, none of the eight repositories provided evidence of such a certification.

In accordance with the Principles of Open Infrastructures, we measured the degree to which the governance of the repository was transparent and documented and whether the repository was stakeholder governed. Only one repository received the highest rating for each of these, while 1 had virtually no information on how the repository is governed, e.g., who is the owner of the repository, or how decisions are made. Although 6 of the repositories were researcher-led, it wasn't always clear how the stakeholder community was involved in oversight, e.g., a scientific advisory board. Finally, the Principles of Open Infrastructures recommends that the software underlying the repository be open source, so that if the repository ceases to be responsive to the community, it could be forked. Two of the repositories provided links to a GitHub repository with a clear open source license.

## Discussion

As part of dkNET.org's efforts to promote data sharing and open science, we undertook an evaluation of current biomedical repositories. Two of the repositories evaluated in this initial round had a dk focus, the Metabolomics Workbench and the AMP-T2D, while the others were either focused on a particular data type, e.g., proteomic, images or flow cytometry, or were generalist repositories that accepted all data types. This study focused primarily on the development and testing of the instrument and the criteria and approach we used for evaluating data repositories. Our intention is to apply it to the entire list of repositories provided by dkNET.

The goal of developing this evaluation instrument was two fold. First, dkNET is developing an on-line catalog of specialist and generalist data repositories suitable for dk researchers to deposit data. The repository catalog is part of a larger effort to help the dk community with FAIR data practices and open science. As one of the most important steps researchers can take towards both open and FAIR is to deposit data into a trustworthy repository that supports both, we wanted to evaluate the extent to which our current ecosystem of biomedical data repositories support practices that are consistent with these principles. Similarly, as researchers may want to receive credit for sharing their data, we wanted to determine the degree to which biomedical repositories supported the requirements for data citation laid out in the data citation principles.

Second, as more attention is now being paid in biomedicine to the services that biomedical repositories should support, including the recently released set of criteria by the National Institutes of Health [16], community organizations like dkNET can serve as a resource both for those maintaining data repositories or developing new ones by helping to define a consistent set of criteria for how dk repositories should operate.

The instrument itself is fairly generic in that many questions would apply to any type of scientific repository. However, we did interpret questions that dealt with community specifics with respect to the needs of biomedicine. For example, we interpreted "rich metadata" as including critical and basic biomedical information such as the type of subject, type of technique and disease condition, that are often left out of general metadata recommendations, e.g., the Data Cite schema, Dublin Core or schema.org. For reusability, we probed for deeper biomedical data, e.g., individual subject attributes such as age and sex, that will promote reuse. As we were specifically interested in open sharing of data, the questionnaire did not contain any questions relating to sensitive data that could not be openly shared, e.g., human subjects information. However, the instrument itself is applicable to all biomedical repositories that have data freely available to the public. We have published the instrument as a FAIR

object under a CC-By license. It is built on open-source software and can be adapted and extended by others.

Overall, as shown in Fig 4, the biomedical repositories we evaluated were considered partially or fully compliant with the OFCT dimensions even though only two of the repositories gave any indication that their functions or design were informed by any of the OFCT principles, in this case specifically mentioning FAIR. The lack of explicit engagement with these principles is not surprising given that most of the repositories were established before these principles came into existence. For this reason, we gave credit for what we called "OFCT potential" rather than strict adherence to a given practice. We used a sliding scale for many questions that would assign partial credit. For example, if the repository did have landing pages at stable URLs we gave them some credit, even if the identifier was not strictly a PID. Such IDs could easily be turned into PIDs by registering them with a resolving service such as Identifiers.org or N2T.org [17]. A good-faith effort was made to try to answer the questions accurately, although reviewing biomedical repositories is challenging. To evaluate specific dimensions required significant engagement with the site, even in some cases requiring us to establish accounts to see what metadata was gathered at time of upload. Discovery of these types of routes, e.g., that ORCIDs are only referenced when you establish an account, required us often to go back and re-evaluate the other repositories using this same method. A follow up with repository managers presenting the result of our review indicated fairly good agreement for most of our evaluations. We acknowledged errors where we missed information that was present on the site. In other cases, we were informed that the repository did support a particular function although no documentation was available through the site. These findings suggest that reviews by independent parties like dkNET can serve a useful function by identifying areas where documentation is missing or unclear or pointing out services that a repository might want to implement, but also point to the importance of verifying any information with repository owners before rendering an evaluation.

In addition to finding relevant information, consistent scoring of the repository was also a challenge. Principles are designed to be aspirational and to provide enough flexibility that they will be applicable across multiple domains. There is therefore a certain amount of subjectivity in their evaluation particularly in the absence of validated, established standards. For example, one of the repositories issued persistent identifiers at the project level but not to the data coming from the individual studies. In another website not included in the final evaluation sample, DOIs were available upon request. Are these considered compliant? One could argue both ways.

As described in the methods, we did not attempt to cover all aspects of the underlying principles, we selected those for which we could develop reasonable evaluation criteria. For example, one very important issue covered by CoreTrustSeal, the newly published TRUST principles [23] and Principles of Open Infrastructure [15] is long term sustainability. Although critical, we do not think that an external party such as ourselves is in a position to comment on the long term sustainability plan for a given repository. Long term sustainability for biomedical infrastructure is a known problem and one for which there are currently few concrete answers as support of most researcher-led infrastructures is in the form of time-limited grants. Our instrument is relevant to this issue, however, as OFCT practices such as FAIR, open formats, open software and good governance practices make repositories more likely to be sustainable as they facilitate transfer of data across organizations.

Since the issuance of the FAIR data principles, several initiatives have invested in the development of tools that are designed to assess the level of data FAIRness, including those that are meant to evaluate on-line data repositories. Some funders such as the EU and NIH are developing policies around FAIR data which may include a more formal assessment of FAIRness.

Such tools include FAIRmetrics (https://github.com/FAIRMetrics/Metrics), FAIR Maturity Indicators [7]), FAIRshake [6] and the FORCE11/Research Data Alliance evaluation criteria [9]. Other efforts, however, extend beyond FAIR and have published lists of functions that should be supported by scientific data repositories in general, e.g., COAR [8], criteria that matter [24] or specifically by biomedical repositories, e.g., NIH [11], Elixir [25]. Our instrument aligns most closely in spirit with COAR, released in October 2020 after this study was completed, as COAR also references the Data Citation Principles, Core Trust Seal and TRUST principles in addition to FAIR.

The FAIRshake toolkits have developed some fully automated or semi-automated approaches for determining FAIRness, e.g., for checking resolvable identifiers. Such tools will make certain of our evaluations easier, e.g., to determine whether machine-readable metadata were available. However, as we show here, some aspects of FAIR require interpretation, e.g., "a plurality of relevant attributes", making it difficult to employ fully automated approaches. In the case of "rich metadata" and "plurality of relevant attributes", dkNET is evaluating these based on our criteria, that is, the type of metadata we think are critical for biomedical studies in our domain. These may not be universal.

While evaluation tools can be powerful, there are downsides to rushing into too rigid an interpretation of any of the principles underlying OFCT, particularly FAIR as that is currently the target of many of these efforts. First, communities are still coming together around good data practices and standards for their constituents based on what can be reasonably implemented at this time. Second, recommendations are being issued by specific stakeholders in the scientific data ecosystem and may not be relevant to other contexts. As noted in the introduction, data repositories have to straddle two worlds: providing traditional publishing/library functions to ensure findability and stability, while at the same fulfilling more traditional roles of scientific infrastructures for harmonizing and reusing data. Thus, evaluating a repository from a journal's or funder's perspective may not be the same as from a researcher's perspective.

Biomedicine also has a very diverse set of repositories, including knowledge bases and sites that are hard to classify as one type or another. One of our sites, AMP-T2D, fell into that category. It is billed as a knowledge portal but provides access to summary statistics from GWAS studies. So although it hosts data, it also has the characteristics of an aggregator site and also a knowledge base. Many of our criteria were hard to interpret in this context and it was a judgement call as to whether all were relevant. However, sites such as AMP = T2D point to the complexity of on-line biomedical data resources. Such results indicate that it is still perhaps early days for understanding what constitutes best practices for a data repository across all disciplines. Our understanding of such practices may evolve over time as data sharing becomes more mainstream. As already noted, for example, early efforts in data sharing necessarily focused on deposition of data. Less attention, perhaps, was paid to what it takes for the effective reuse of the data. While the FAIR principles emphasize machine-readable attributes for achieving reusability without human intervention, some studies suggest that the human factor may be more critical for some types of data [26] For these types, having a contact person and an accompanying publication makes it much easier to understand key contextual details [26,27]. As we start to see more reuse of data, it may be possible to employ more analytical methods for determining best practices based on actual use cases.

For these reasons, we deliberately refrained from assigning grades or calling out individual repositories in the work presented here. [7] noted that many repositories which were evaluated early on using FAIRmetrics expressed resentment. We recognize the struggles that those who develop and host scientific data repositories undergo to keep the resource up and running, particularly in the face of uncertain funding. Generally, these repositories were founded to

serve a particular community, and the community itself may not be demanding or engaging with OFCT principles. We therefore favor flexible approaches that allow individual communities to interpret OFCT within the norms of their community and not entirely according to the dictates of external evaluators. Nevertheless, research data repositories, after operating largely on their own to determine the best way to serve research data, are going to have to adapt to meet the challenges and opportunities of making research data a primary product of scientific research.

## Supporting information

**S1 Table. Question identifiers and their abbreviated meaning. PolicyModels allows specifying an id for each question. This id is later used to identify that question, and to localize its text.** The ids we use here pertain to the subject of the question. The table below explains each abbreviation.
(DOCX)

**S2 Table. Abbreviations used in text.**
(DOCX)

## Acknowledgments

The authors wish to thank Drs. Ko Wei Lin and Jeffrey Grethe for helpful comments.

## Author Contributions

**Conceptualization:** Fiona Murphy, Maryann E. Martone.

**Data curation:** Fiona Murphy, Michael Bar-Sinai, Maryann E. Martone.

**Formal analysis:** Fiona Murphy, Michael Bar-Sinai, Maryann E. Martone.

**Funding acquisition:** Maryann E. Martone.

**Investigation:** Fiona Murphy, Michael Bar-Sinai.

**Methodology:** Fiona Murphy, Michael Bar-Sinai, Maryann E. Martone.

**Project administration:** Maryann E. Martone.

**Resources:** Maryann E. Martone.

**Software:** Michael Bar-Sinai.

**Supervision:** Maryann E. Martone.

**Validation:** Michael Bar-Sinai.

**Visualization:** Michael Bar-Sinai.

**Writing – original draft:** Fiona Murphy, Michael Bar-Sinai, Maryann E. Martone.

**Writing – review & editing:** Fiona Murphy, Michael Bar-Sinai, Maryann E. Martone.

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
