## [Decision Letter · Decision Letter 0]

27 Jan 2021

PONE-D-20-39290

Alignment of biomedical data repositories with open, FAIR, citable and trustworthy principles

PLOS ONE

Dear Dr. Martone,

Thank you for submitting your manuscript to PLOS ONE. After careful consideration, we feel that it has merit but does not fully meet PLOS ONE’s publication criteria as it currently stands. Therefore, we invite you to submit a revised version of the manuscript that addresses the points raised during the review process.

**I have a few specific comments **: 

- Please state if the research was pre-registered or not explicitely in the method section ; 

- Figure 2 would be better in a supplementary material (in my opinion) ;

- On Figures 5 and 6, please provide the name of each repository as a legend (if possible) ; 

We look forward to receiving your revised manuscript.

Kind regards,

Florian Naudet, M.D., M.P.H., Ph.D.

Academic Editor

PLOS ONE

Journal Requirements:

"This work was supported by NIH grant# 3U24DK097771-08S1 from the National Institutes of Diabetes and Digestive and Kidney Diseases.  The original decision tree was developed through the  FORCE11 Scholarly Commons working group supported by an award from The Leona M. and Harry B. Helmsley Charitable Trust Biomedical Research Infrastructure Program to FORCE11.  The authors wish to thank Drs. Ko Wei Lin and Jeffrey Grethe for helpful comments.  The funders played no role in the design, data collection, analysis, decision to publish or preparation of the manuscript."

We note that one or more of the authors are employed by a commercial company: MoreBrains Cooperative Ltd, Chichester, UK

"I have read the journal's policy and the authors of this manuscript have the following competing interests: Dr. Martone is on the board and has equity interest in SciCrunch Inc., a tech startup that develops tools and services in support of Research Resource Identifiers.

Dr Murphy is on the board of Dryad Data Repository."

Please respond by return email with your amended Competing Interests Statement and we will change the online submission form on your behalf.

Reviewers' comments:

Reviewer's Responses to Questions

**Comments to the Author**

1. Is the manuscript technically sound, and do the data support the conclusions?

Reviewer #1: Yes

Reviewer #2: Partly

Reviewer #3: Yes

2. Has the statistical analysis been performed appropriately and rigorously? 

Reviewer #1: Yes

Reviewer #2: N/A

Reviewer #3: I Don't Know

3. Have the authors made all data underlying the findings in their manuscript fully available?

Reviewer #1: Yes

Reviewer #2: Yes

Reviewer #3: Yes

4. Is the manuscript presented in an intelligible fashion and written in standard English?

Reviewer #1: Yes

Reviewer #2: Yes

Reviewer #3: Yes

5. Review Comments to the Author

Reviewer #1: First of all, I want to congratulate the authors on this interesting piece of research.

In times of open data, data repositories are evolving rapidly and thus, it is even more important to have scientific articles analyzing them.

A further plus of this paper is the openness and transparency which is shown by sharing the code and the data. Finally, the paper shows a well thought through study design.

However, there are some points that need to be improved.

I recommend an acceptance with minor improvements.

As the lines are not numbered, I will go through the paper section by section.

Title:

Please include the design of the research in the title and its outcome

Abstract:

Please include that you are looking for repositories focusing on diabetes and metabolic diseases.

Again, the design of the study is missing

Please include the time frame of when the research was conducted.

Introduction:

Overall, I am missing references to already established research, but particularly in the introduction section. Was this the only research initiative on NIH repositories.

Methods:

Please explain how many repositories are on the dkNet website and describe the outsourcing process. A flowchart could be useful here for readers.

Did you preregister a protocol on OSF, etc? If so, please insert the link. If not, please explain why.

Did you pilot test the questionnaire on other repositories?

Please include references for the TRUST principles and the CoreTrust Seal.

Table 2 is kind of confusing with the Q#ID for the reader. In my opinion not necessary as it is in the supplementary material S1 again listed. However, personal preference here.

Same problem in Table 3.

Is listing RRID necessary?

What is the point of the section “section”. For the AMP-T2D you put data. My understanding is that “data” would be applicable to all repositories.

Please take care that you explain abbreviations. These might not always be clear for readers who are not familiar with the subject (RRID, API, …). It would be good to have a list of abbreviations in the beginning of the manuscript.

Please state if you followed a reporting guideline, STROBE would be a possibility here. If you don’t use one, please explain why.

Results:

When you finished the research and didn’t find certain items, did you then contact the responsible or repository managers?

Keep in mind that it is highly appreciated that you created accounts in some cases.

I am sorry if I didn’t understand this point but how did you get to a 10 point scale if in some questions you had only 3 or 4 options to choose.

On page 16 you are referring to table 3 when you mean table 2.

Discussion:

I am again missing the focus on the specific disease area. You are writing in a general way on repositories but in which way do your results have external validity for other biomedical areas?

The study design is very well done, however only applied on 8 repositories; should be further discussed/ mentioned.

Reviewer #2: The authors present a framework for data repositories to support Open, FAIR, Citable and Trustworthy (OFCT) data. They describe the application of the open source PolicyModels toolkit to operationalize key aspects of OFCT principles. They use the framework to assess the current state of eight biomedical community repositories listed by the NIDDK Information Network (dkNET). The dkNET is taking an active role in interpreting and facilitating compliance with FAIR on behalf of this community. By defining a set of important functions repositories should support, the work will support two goals: (i) serve as a best practice for those developing new dk data repositories, and (ii) inform the creation of a dkNET tool to help researchers select an appropriate repository for their data.

In general, assessment of repositories and is an important area, and more attention is being paid in biomedicine to certification instruments. I like very much the online PolicyModels toolkit, its functionality and usability, and the machine-readable exports.

While I applaud the goals, in general, I have the following concerns.

1. The process, and the rationale for the development of the OFCT principles are not explained. As the authors say “To our knowledge, this is the first evaluation instrument that was designed specifically around OFCT. However, since the issuance of the FAIR data principles, several initiatives have invested in the development of tools that are designed to assess the level of data FAIRness, including those that are meant to evaluate on-line data repositories”. To my knowledge, beside FAIR and the CTS certifications, there are the several community-driven ‘principles’ to assess repositories and/or provide best pratices for repositories developers and maintainers, by the following groups: COAR (https://www.coar-repositories.org/coar-community-framework-for-good-practices-in-repositories), NIH (https://grants.nih.gov/grants/guide/notice-files/NOT-OD-21-016.html), ELIXIR (https://doi.org/10.12688/f1000research.9656.2) and a group of publishers (https://doi.org/10.5281/zenodo.4084763). To address this, I would like to see the authors explaining: (i) how OFCT was developed, in terms of community participation, and how the range of acceptable and unacceptable values in various dimensions (properties and flags) were defined; (ii) the rationale about creating these new set of principles, and how OFCT is different, complementary or more usable than the others; and (i) how the OFCT maps to these existing efforts, and to the CTS.

2. Although FAIR is only one of the elements of the OFCT principles, much of the discussion is only centered on comparing them to FAIR assessment tools, such as the FAIR Maturity Indicators, FAIR Evaluator and FAIRshake, which are solely based on FAIRness. To address this, I would like to see the authors expanding the discussion to on the rest of the elements in the OFCT principles, which can be done by addressing my point 1.

3. It seems that the application of the OFCT principles is based on manual assessment and on the interpretation of the results. The authors say: “…we deliberately refrained from assigning grades or calling out individual repositories in the work presented here. (Wilkinson et al. 2019) noted that many repositories which were evaluated early on using FAIRmetrics expressed resentment. We recognize the struggles that those who develop and host scientific data repositories undergo to keep the resource up and running, particularly in the face of uncertain funding……We therefore favor flexible approaches that allow individual communities to interpret OFCT within the norms of their community and not entirely according to the dictates of external evaluators.” And also: “The FAIR Maturity Indicators and FAIRshake toolkits differ from ours in that they are intended to employ either fully automated or semi-automated approaches for determining FAIRness. As we show here, some aspects of FAIR require interpretation, e.g., “a plurality of relevant attributes”, making it difficult to employ fully automated approaches”. I understand that fully automated approaches may not be easy, but I do not see how manual assessment would be better and work in practice, in particular be objective and scalable. To address this, I would like to see the authors explaining how they see the manual assessment (with OFCT principles), and what they call the “OFCT potential”, to help them to develop a tool to select an appropriate repository (one of their goals).

Other comments

4. The assessment was performed through inspection of documentation and interaction with the sites, without interacting with the repository’s owners. However, this is not clear. Fig 2 talks about “survey and questionnaire and interactive interview”, but then in the Scoring section authors say : “Five of the sites were reviewed independently by FM and MM…The reviewers made a good faith effort to find information on the site to provide an accurate answer for each question”. Therefore, the authors should make clear that this not a survey sent to repository’s owners or a self-assessment done by the repository’s owners. Nevertheless, the authors could contact the repository’s owners to vet the results.

5. The authors say that they have benchmarked their instrument against others for ‘similar use’, such as: the Datacite’s Repository Finders, and the Scientific Data’s Repository Questionnaire, and the FAIRsFAIR assessment tool. However, there is not information on what this benchmarked has produced since these efforts are not equivalent and the OFCT are principles and indeed not a tool.

6. The data is well stored and cited, as expected, but there are a lot of key details scattered across figures and in the GitHub repositories, and in places it is quite hard to follow.

Reviewer #3: The paper and study have were clearly set out. In particular, the discussion provided insight into the difficulty of other toolkits, limitations on a universal approach and how this study addressed these challenges.

The paper could have included more detail about how the toolkit will impact the workflows of researchers (for the better) and how the community can apply the toolkit in different contexts. It is not immediately clear how the toolkit operationalises key aspects of OFCT principles and this could be further highlighted. It should be noted, however, the toolkit and related materials have been made available for reuse.

It was an interesting paper that has built on work that has been done by other groups and provided a good framework for the community to adopt.

6. PLOS authors have the option to publish the peer review history of their article (what does this mean?). If published, this will include your full peer review and any attached files.

Reviewer #1: No

Reviewer #2: No

Reviewer #3: No

---

## [Author Response · Author response to Decision Letter 0]

7 Apr 2021

We responded to all comments raised by the editor and reviewers. I uploaded a response to the review but I'm including them here as well. 

Response to reviewer comments

We thank the editor and the reviewers for their thoughtful and helpful comments. We have revised the manuscript substantially in response to these critiques. 

Editor’s comments

PO-1: Attached - marked-up copy of the manuscript labelled 'Revised Manuscript with Track Changes', and clean version labelled ‘Manuscript’

Response: We have attached both a version with track changes on and a clean version labeled as per above.

PO-2: Please state if the research was pre-registered or not explicitly in the method section ; 

Response: Done

PO-3: Figure 2 would be better in a supplementary material (in my opinion) ;

Reponse: Would prefer to leave in main text as one of our audience targets is repository owners or evaluators who may want to use the instrument. So we wanted to show them the user-friendliness of the user interface. However, if the editor prefers, we can move it. 

PO-4: On Figures 5 and 6, please provide the name of each repository as a legend (if possible) ; 

Response: We prefer not to identify individual repositories in the manuscript, as our intent is to examine overall trends and practices, not to critique individual repositories. The scores for individual repositories are, however, available in the data provided.

PO-5: If applicable, we recommend that you deposit your laboratory protocols in protocols.io to enhance the reproducibility of your results. Protocols.io assigns your protocol its own identifier (DOI) so that it can be cited independently in the future.

Response: We don’t feel it necessary as the protocol is documented in the instrument and the particular instrument used was given a DOI with a clear version number. 

Journal Requirements:

"This work was supported by NIH grant# 3U24DK097771-08S1 from the National Institutes of Diabetes and Digestive and Kidney Diseases. The original decision tree was developed through the FORCE11 Scholarly Commons working group supported by an award from The Leona M. and Harry B. Helmsley Charitable Trust Biomedical Research Infrastructure Program to FORCE11. The authors wish to thank Drs. Ko Wei Lin and Jeffrey Grethe for helpful comments. The funders played no role in the design, data collection, analysis, decision to publish or preparation of the manuscript. 

We note that one or more of the authors are employed by a commercial company: MoreBrains Cooperative Ltd, Chichester, UK

Response: We appended the following statement to our Funding statement: “ Fiona Murphy is employed by MoreBrains Cooperative Ltd, Chichester, UK, a commercial company that did not provide any direct funding for this project and does not have any direct interests in the repositories, dkNET or NIH."

 Response: Done

Response: Done

"I have read the journal's policy and the authors of this manuscript have the following competing interests: Dr. Martone is on the board and has equity interest in SciCrunch Inc., a tech startup that develops tools and services in support of Research Resource Identifiers.

Dr Murphy is on the board of Dryad Data Repository."

Response: Done

Please respond by return email with your amended Competing Interests Statement and we will change the online submission form on your behalf.

Reviewer #1

Reviewer #1: First of all, I want to congratulate the authors on this interesting piece of research.

In times of open data, data repositories are evolving rapidly and thus, it is even more important to have scientific articles analyzing them.A further plus of this paper is the openness and transparency which is shown by sharing the code and the data. Finally, the paper shows a well thought through study design.

Response: We thank the reviewer for these positive comments

1-1: Title: Please include the design of the research in the title and its outcome

Response: We modified the title to: “A new tool for assessing aAlignment of biomedical data repositories with open, FAIR, citable and trustworthy principles”

1-2: Abstract: Please include that you are looking for repositories focusing on diabetes and metabolic diseases. Again, the design of the study is missing. Please include the time frame of when the research was conducted.

Response: As only two of the repositories were specifically focused on diabetes and metabolomics, we added the following statement: Repositories included both specialist repositories that focused on a particular data type or domain, in this case diabetes and metabolomics, and generalist repositories that accept all data types and domains. 

We also added some additional methodological details and the timeline over which the repositories were evaluated.

1-3: Introduction: Overall, I am missing references to already established research, but particularly in the introduction section. Was this the only research initiative on NIH repositories.

Response: We included references to other instruments that have been designed to evaluate scientific and biomedical data repositories and added the following text:

Because of the central importance of data repositories to achieving goals in open science and FAIR, many groups have been developing criteria and instruments to assess the state of scientific data repositories, including biomedical repositories, e.g., FAIRmetrics, FAIRshake, Criteria that matter, to name a few. In the work presented here, we wanted to expand the evaluation criteria based on OFCT and to specifically assess repositories that are relevant to a specific domain, diabetes, digestive and kidney diseases, on developed an instrument to assess the current state of data repositories on behalf of the NIDDK Information Network (dkNET.org)

1-4: Methods:

Please explain how many repositories are on the dkNet website and describe the outsourcing process. A flowchart could be useful here for readers.

Response: There are over 80 repositories listed on the dkNET website. We are not sure what is meant by “outsourced” here. We added more information to the manuscript about how dkNET chooses repositories for listing. We explain in the manuscript how we chose a sample for evaluation. 

Did you preregister a protocol on OSF, etc? If so, please insert the link. If not, please explain why.

Response: No, we did not and we have added a statement to the manuscript to that effect. However, a preprint was posted in BiorXiv. We view that as a form of community review prior to publication. However, it is a good suggestion to register all such studies in the future and we appreciate you bringing it to our attention. 

Did you pilot test the questionnaire on other repositories?

Response: Yes, we tested the questionnaire on 13 repositories initially to pilot the questionnaire. Some of them are the same repositories that we evaluated for the final sample, but not all repositories evaluated were in the initial set. 

Please include references for the TRUST principles and the CoreTrust Seal.

Response: Both references to the Core Trust Seal and the TRUST principles are included.

Table 2 is kind of confusing with the Q#ID for the reader. In my opinion not necessary as it is in the supplementary material S1 again listed. However, personal preference here.

Response: Agreed. We removed the IDs but left in the question number for ease of reference in the text. 

Same problem in Table 3.

Is listing RRID necessary?

Response: Yes, the RRID is used to make it easy to track use of repositories and is supported by PLoS. However, we removed the column entitled “RRID” and just listed it under each repository in smaller font so it isn’t so prominent.

What is the point of the section “section”. For the AMP-T2D you put data. My understanding is that “data” would be applicable to all repositories.

Response: Many of the websites are complicated scientific applications with multiple sections. We wanted to be clear about which section we reviewed for compliance. In the AMP-T2D, the tab heading is entitled “Data” and that is what we referred to. However, we modified the text to indicate the title of the page and the section header to make it more clear. 

Please take care that you explain abbreviations. These might not always be clear for readers who are not familiar with the subject (RRID, API, …). It would be good to have a list of abbreviations in the beginning of the manuscript.

Response: Done. We included it in the supplemental materials. 

Please state if you followed a reporting guideline, STROBE would be a possibility here. If you don’t use one, please explain why.

Response: We followed the guidelines recommended for FAIR metrics (https://www.nature.com/articles/sdata2018118/tables/1

1-5: Results:

When you finished the research and didn’t find certain items, did you then contact the responsible or repository managers?

Keep in mind that it is highly appreciated that you created accounts in some cases.

Response: We did not but in response to this comment, we have sent the completed questionnaires to all of the repositories and have described the results in the text. At the time of deadline for resubmission, we received 3 responses. If by the time of rereview we receive more, we will update accordingly.

I am sorry if I didn’t understand this point but how did you get to a 10 point scale if in some questions you had only 3 or 4 options to choose.

Response: Questions could have 2, 3 or 4 possible answers so to make it easier to compare across questions to gauge the overall alignment with each principle , we normalized all questions to a 10 point scale. We added a sentence to the text to explain.

On page 16 you are referring to table 3 when you mean table 2.

Response: As we removed the IDs from that table, we removed the erroneous reference to Table 3 as it was no longer relevant.

1-6: Discussion:

I am again missing the focus on the specific disease area. You are writing in a general way on repositories but in which way do your results have external validity for other biomedical areas?

Response: We added a paragraph to the discussion on this point. While the majority of questions would apply to all scientific data repositories, for questions that probed the level of metadata, we specifically looked for metadata relevant for biomedicine.

The study design is very well done, however only applied on 8 repositories; should be further discussed/ mentioned.

Response: We mention that this study focused on the design of the instrument and an initial evaluation of eight representative repositories to see how it performed. We intend to apply it to all repositories listed in dkNET. 

Reviewer #2:

Reviewer #2: The authors present a framework for data repositories to support Open, FAIR, Citable and Trustworthy (OFCT) data. They describe the application of the open source PolicyModels toolkit to operationalize key aspects of OFCT principles. They use the framework to assess the current state of eight biomedical community repositories listed by the NIDDK Information Network (dkNET). The dkNET is taking an active role in interpreting and facilitating compliance with FAIR on behalf of this community. By defining a set of important functions repositories should support, the work will support two goals: (i) serve as a best practice for those developing new dk data repositories, and (ii) inform the creation of a dkNET tool to help researchers select an appropriate repository for their data.

In general, assessment of repositories and is an important area, and more attention is being paid in biomedicine to certification instruments. I like very much the online PolicyModels toolkit, its functionality and usability, and the machine-readable exports.

Response: We thank the reviewer for the positive comments

While I applaud the goals, in general, I have the following concerns.

2-1. The process, and the rationale for the development of the OFCT principles are not explained. As the authors say “To our knowledge, this is the first evaluation instrument that was designed specifically around OFCT. However, since the issuance of the FAIR data principles, several initiatives have invested in the development of tools that are designed to assess the level of data FAIRness, including those that are meant to evaluate on-line data repositories”. To my knowledge, beside FAIR and the CTS certifications, there are the several community-driven ‘principles’ to assess repositories and/or provide best practices for repositories developers and maintainers, by the following groups: COAR (https://www.coar-repositories.org/coar-community-framework-for-good-practices-in-repositories), NIH (https://grants.nih.gov/grants/guide/notice-files/NOT-OD-21-016.html), ELIXIR (https://doi.org/10.12688/f1000research.9656.2) and a group of publishers (https://doi.org/10.5281/zenodo.4084763). To address this, I would like to see the authors explaining: (i) how OFCT was developed, in terms of community participation, and how the range of acceptable and unacceptable values in various dimensions (properties and flags) were defined; (ii) the rationale about creating these new set of principles, and how OFCT is different, complementary or more usable than the others; and (i) how the OFCT maps to these existing efforts, and to the CTS.

Response: We added additional background material in the introduction that explains the origins of the OFCT framework through the work of FORCE11. We note that the principles referenced in OFCT were not developed by us but rather through large community efforts, e.g., the Data Citation Principles, FAIR and Trust, or third parties (Principles of open infrastructure and the Open Definition. We arrived at OFCT through a process of community workshops in FORCE11 and through analysis of dozens of principle sets issued around the world that emphasized openness, reusability, credit and trustworthiness. We operationalized these concepts first by selecting community principles that embody these concepts and then selecting criteria through which they could be evaluated. In addition, we add references to COAR, Elixir and NIH which were missing from our article. We note that COAR, which was released in Oct 2020 after our instrument was complete actually is close in spirit to OFCT and therefore complementary to our efforts. We include a discussion of that point. 

2-2. Although FAIR is only one of the elements of the OFCT principles, much of the discussion is only centered on comparing them to FAIR assessment tools, such as the FAIR Maturity Indicators, FAIR Evaluator and FAIRshake, which are solely based on FAIRness. To address this, I would like to see the authors expanding the discussion to on the rest of the elements in the OFCT principles, which can be done by addressing my point 1.

Response: We had referenced some of these other efforts in the discussion, but agree that they should be brought up earlier, so we have added more background to the introduction to reference additional efforts. 

2-3. It seems that the application of the OFCT principles is based on manual assessment and on the interpretation of the results. The authors say: “…we deliberately refrained from assigning grades or calling out individual repositories in the work presented here. (Wilkinson et al. 2019) noted that many repositories which were evaluated early on using FAIRmetrics expressed resentment. We recognize the struggles that those who develop and host scientific data repositories undergo to keep the resource up and running, particularly in the face of uncertain funding……We therefore favor flexible approaches that allow individual communities to interpret OFCT within the norms of their community and not entirely according to the dictates of external evaluators.” And also: “The FAIR Maturity Indicators and FAIRshake toolkits differ from ours in that they are intended to employ either fully automated or semi-automated approaches for determining FAIRness. As we show here, some aspects of FAIR require interpretation, e.g., “a plurality of relevant attributes”, making it difficult to employ fully automated approaches”. I understand that fully automated approaches may not be easy, but I do not see how manual assessment would be better and work in practice, in particular be objective and scalable. To address this, I would like to see the authors explaining how they see the manual assessment (with OFCT principles), and what they call the “OFCT potential”, to help them to develop a tool to select an appropriate repository (one of their goals).

Response: We agree that we did not tie the two themes together effectively and have not added more material as to how the manual evaluation helps an organization like dkNET to improve the quality of repositories serving their domains and provide useful information to our constituents. We have strengthened the basic premise that while repositories should have a set of core functions they support, e.g., PIDs, different groups may evaluate their fitness of purpose differently. 

Other comments

2-4. The assessment was performed through inspection of documentation and interaction with the sites, without interacting with the repository’s owners. However, this is not clear. Fig 2 talks about “survey and questionnaire and interactive interview”, but then in the Scoring section authors say : “Five of the sites were reviewed independently by FM and MM…The reviewers made a good faith effort to find information on the site to provide an accurate answer for each question”. Therefore, the authors should make clear that this not a survey sent to repository’s owners or a self-assessment done by the repository’s owners. Nevertheless, the authors could contact the repository’s owners to vet the results.

Response: We have made that point clearer in the methods. In addition, following the reviewers’ recommendation, we sent the results of our assessment to the 8 repositories. We had acknowledgements from 7 of them and received feedback from 3 of them to date. We have included this in our methods and results. 

2-5. The authors say that they have benchmarked their instrument against others for ‘similar use’, such as: the Datacite’s Repository Finders, and the Scientific Data’s Repository Questionnaire, and the FAIRsFAIR assessment tool. However, there is not information on what this benchmarked has produced since these efforts are not equivalent and the OFCT are principles and indeed not a tool.

Response: We have expanded on the explanation for this section in the methods section to explain that we were seeking both context for our own work and information as to what is required for a thorough questionnaire that will be of use to its potential community.

2-6. The data is well stored and cited, as expected, but there are a lot of key details scattered across figures and in the GitHub repositories, and in places it is quite hard to follow.

Response: We tried to simplify the figures and the amount of cross referencing that is necessary. We have also deposited all of the code and data underlying this study in Zenodo and we make that clear in the first paragraph of the data availability statement. 

Reviewer #3

Reviewer #3: The paper and study were clearly set out. In particular, the discussion provided insight into the difficulty of other toolkits, limitations on a universal approach and how this study addressed these challenges.

Response: We thank the reviewer for his positive comments

3-1. The paper could have included more detail about how the toolkit will impact the workflows of researchers (for the better) and how the community can apply the toolkit in different contexts. It is not immediately clear how the toolkit operationalises key aspects of OFCT principles and this could be further highlighted. It should be noted, however, the toolkit and related materials have been made available for reuse.

Response: We added additional detail about how this instrument will be used in dkNET and can be used in different contexts. 

3-2. It was an interesting paper that has built on work that has been done by other groups and provided a good framework for the community to adopt.

Thank you for the positive comment.

---

## [Decision Letter · Decision Letter 1]

8 Jun 2021

A tool for assessing alignment of biomedical data repositories with open, FAIR, citation and trustworthy principles

PONE-D-20-39290R1

Dear Dr. Martone,

We’re pleased to inform you that your manuscript has been judged scientifically suitable for publication and will be formally accepted for publication once it meets all outstanding technical requirements.

Kind regards,

Florian Naudet, M.D., M.P.H., Ph.D.

Academic Editor

PLOS ONE

Additional Editor Comments (optional):

**I would like to thank the 3 reviewers for their comments and for pointing that the manuscript is in a better shape now. I'm happy to accept it for publication. **

**As I have informed the authors, it was not possible for one of the reviewer to provide a fast answer. He informed me appropriately and I have decided to give him more time because I wanted to have his opinion on the revised manuscript. I apologize for the time taken but it was very important for me to have his opinion on this new version. I hope that the authors will understand. **

Reviewers' comments:

Reviewer's Responses to Questions

**Comments to the Author**

1. If the authors have adequately addressed your comments raised in a previous round of review and you feel that this manuscript is now acceptable for publication, you may indicate that here to bypass the “Comments to the Author” section, enter your conflict of interest statement in the “Confidential to Editor” section, and submit your "Accept" recommendation.

Reviewer #1: All comments have been addressed

Reviewer #2: All comments have been addressed

Reviewer #3: All comments have been addressed

2. Is the manuscript technically sound, and do the data support the conclusions?

Reviewer #1: Yes

Reviewer #2: Yes

Reviewer #3: Yes

3. Has the statistical analysis been performed appropriately and rigorously? 

Reviewer #1: Yes

Reviewer #2: N/A

Reviewer #3: I Don't Know

4. Have the authors made all data underlying the findings in their manuscript fully available?

Reviewer #1: Yes

Reviewer #2: Yes

Reviewer #3: Yes

5. Is the manuscript presented in an intelligible fashion and written in standard English?

Reviewer #1: Yes

Reviewer #2: Yes

Reviewer #3: Yes

6. Review Comments to the Author

Reviewer #1: I thank the authors for addressing all my comments. This is a nice piece of work and I further wish them a smooth publication process.

Reviewer #2: I thank the authors for their thorough response and revision of the manuscript, which addresses my comments in a satisfactory manner.

Reviewer #3: The revised submission has adequately addressed my comments raised previously. There is growing interest in the assessment of repositories globally and the authors have developed a framework and tool that can be adapted across other domains. The authors have provided a comprehensive overview of the current biomedical repository assessment landscape and developed an instrument that goes beyond the FAIR principles. This is an important step in the Open Science community as repositories need to adapt to best practice. The toolkit can be used for repositories to benchmark against best practice as they continue to evolve. It would have been good to see more about the exploitation or proposed uptake of the toolkit, beyond making the code openly available, although this does not detract from the merit of the paper.

7. PLOS authors have the option to publish the peer review history of their article (what does this mean?). If published, this will include your full peer review and any attached files.

Reviewer #1: No

Reviewer #2: **Yes: **Susanna-Assunta Sansone

Reviewer #3: No

---

## [Editor Report · Acceptance letter]

28 Jun 2021

PONE-D-20-39290R1 

A tool for assessing alignment of biomedical data repositories with open, FAIR, citation and trustworthy principles 

Dear Dr. Martone:

I'm pleased to inform you that your manuscript has been deemed suitable for publication in PLOS ONE. Congratulations! Your manuscript is now with our production department. 

Kind regards, 

on behalf of

Pr. Florian Naudet 

Academic Editor

PLOS ONE